# Multi-omics reveal neuroprotection of *Acer truncatum Bunge* Seed extract on hypoxic-ischemia encephalopathy rats under high-altitude

Xianyang Chen[1,7], Yige Song[1,7], Wangting Song[1,7], Jiarui Han[1,7], Hongli Cao[2,7], Xiao Xu[3], Shujia Li[3], Yanmin Fu[3], Chunguang Ding[4], Feng Lin[5], Yuan Shi [6] & Jiujun Li [3✉]

Hypoxic-ischemic encephalopathy (HIE) at high-altitudes leads to neonatal mortality and long-term neurological complications without effective treatment. *Acer truncatum Bunge* Seed extract (ASO) is reported to have effect on cognitive improvement, but its molecular mechanisms on HIE are unclear. In this study, ASO administration contributed to reduced neuronal cell edema and improved motor ability in HIE rats at a simulated 4500-meter altitude. Transcriptomics and WGCNA analysis showed genes associated with lipid bio-synthesis, redox homeostasis, neuronal growth, and synaptic plasticity regulated in the ASO group. Targeted and untargeted-lipidomics revealed decreased free fatty acids and increased phospholipids with favorable ω-3/ω-6/ω-9 fatty acid ratios, as well as reduced oxidized glycerophospholipids (OxGPs) in the ASO group. Combining multi-omics analysis demonstrated FA to FA-CoA, phospholipids metabolism, and lipid peroxidation were regulated by ASO treatment. Our results illuminated preliminary metabolism mechanism of ASO ingesting in rats, implying ASO administration as potential intervention strategy for HIE under high-altitude.

[1] Bao Feng Key Laboratory of Genetics and Metabolism, Beijing, China. [2] Department of Respiratory, Beijing Rehabilitation Hospital, Capital Medical University, Beijing, China. [3] Department of Pediatrics, Shengjing Hospital of China Medical University, Plateau Medical Research Center of China Medical University, Shenyang, China. [4] National Center for Occupational Safety and Health, Beijing, China. [5] Department of Neurology, Sanming First Hospital Affiliated to Fujian Medical University, Sanming, Fujian, China. [6] Department of Neonatology, Children's Hospital Affiliated Chongqing Medical University, Chongqing, China. [7] These authors contributed equally: Xianyang Chen, Yige Song, Wangting Song, Jiarui Han, Hongli Cao. ✉email: lijj@sj-hospital.org

Hypoxic-ischemic encephalopathy (HIE) represents a multifaceted pathological progression induced by insufficient oxygen delivery to the central nervous system (CNS). This leads to disturbances in metabolism, cerebral edema, and neuronal cell death[1], culminating in impairments to cognitive and motor function[2]. In high-altitude regions, residents, in particular, newborns face a significant health peril in the form of HIE, owing to reduce atmospheric pressure and other associated factors[3]. While various medications can mitigate intracranial pressure and alleviate patient symptoms, the treatment outcomes for neuronal damage caused by HIE remain unsatisfactory[4]. Consequently, there is an urgent need for effective interventions to address this condition.

Fatty acids (FAs), including the unsaturated fatty acids (UFAs) found in Omega-3, Omega-6 and Omega-9 series, have been proposed as having substantial potential in the prevention and treatment of HIE[5–7]. Recent research has found that supplementation with these FAs can not only improve brain function, but also enhance memory, learning, and cognitive abilities[8–11]. However, it is only in recent years that extensive research has been conducted on the neuroprotective effects of FAs under hypoxic-ischemic conditions. For example, Zhang's study revealed that FAs have potent neuroprotective effects against hypoxic-ischemic injury[12]. Furthermore, supplementation with fish oil has been reported to reduce neuronal inflammation and depression-like behavior[13]. These findings provide new evidence of the potential of FAs supplementation as a therapeutic strategy for HIE and associated neurological conditions.

*Acer truncatum Bunge* seed extract (ASO) is an important woody oil, of which nutritional and economic value predominantly depends on its fatty acid composition. Notably, ASO serves as a primary botanical source for the large-scale production of nervonic acid (C24:1, Δ15, cis-15-Tetracosenoic acid, NA), constituting 3–7% of its composition[8]. Previous reports have documented the capacity of ASO supplementation to effectively improve the cognitive abilities of rats[8], and participants in the diet therapy for mice model of multiple sclerosis[14]. The Omega-3, Omega-6, and Omega-9 FAs in ASO, in addition to regulating the immune system, have anti-inflammatory and antioxidant properties, which play an important role in maintaining normal neural transmission, protecting nerve cells, and promoting nerve regeneration[15–17]. Our previous research revealed that the administration of ASO significantly improved the learning and memory capabilities of rats, and regulated the rebalance of ω-3/ω-6/ω-9 in the body, indicating its potential drug development as well as a natural source of structural-NA[8].

In this study, we employed an oxygen chamber to simulate the high-altitude conditions of Nagqu, Tibet, situated at an altitude of 4500 meters. Our primary objective was to evaluate the effect of ASO as an intervention for alleviating brain injuries in neonates affected by HIE exposed to high-altitude conditions. To explore the mechanism of ASO intervention, we conducted a comprehensive analysis of lipidomics and transcriptomics, using corn oil as the control in this investigation[18,19]. Our results may provide insights into the molecular mechanisms of ASO and its potential to protect the nervous system and improve motor and cognitive abilities in neonates with HIE.

## Results
### ASO improved autonomic motor ability in HIE rats.
The experimental procedure was shown in Fig. 1a. We evaluated the intervention effect of ASO administration on HIE rats exposed to high-altitude environments through comprehensive investigations involving pathology, behavior test, lipidome and transcriptome. HE staining and TTC enabled pathological evaluation

of the Sham group, Model group, Control group, and ASO group. HE staining revealed that the nerve cells in the brain tissue of the Sham group were neatly arranged, the nuclei were visible, the nucleoli were clear, and the cytoplasm was abundant, whereas in the Model group, nerve cells exhibited disordered and neuronal edema (Supplementary Fig. 1b, d). The damaged areas in the figure were marked, and the results showed that the volume ratio of the ischemic area stained by TTC in the Model group was 12.84% (Supplementary Fig. 1a).

After 30 days of intervention, ASO group underwent intragastric ASO treatment, whereas HIE rats in the Control group received an equivalent volume of corn oil as the control vehicle. The brain tissue damage in the ASO group was lighter, the cell arrangement was restored neatly, and the neurons were slightly edematous compared with the Control group, indicating that ASO alleviated the brain damage caused by HIE in the high-altitude environment (Fig. 1b, d). The results of the open field test displayed that the performance of distance moved and duration in the center in ASO group was significantly more effective than the Control group, suggesting that the autonomic motor ability of mice was significantly improved after taking ASO (Fig. 1c, e, f). However, no significant differences in movement duration were observed between the Control group and the ASO group in the open field test (Supplementary Fig. 1c).

### Transcriptome analysis of brains in both ASO and Control groups.
Transcriptome analysis was conducted on rat brain tissue. Sample detection showed excellent reproducibility (Fig. 2a). Differential multiple fold change (FC) ≥ 2 or FC ≤ 0.5 (the absolute value of log2 FC ≥ 1) and $Q$-value ≤ 0.05 as the criterion, the 249 genes were screened as differentially expressed genes (DEGs). Among them, 187 of DEGs were up-regulated, while 62 were down-regulated (Fig. 2b). The heatmap displayed the expression of genes in different samples (or different treatments), and cluster analysis indicated the two groups were distinguished by transcriptome (Fig. 2c). The enrichment analysis indicated that the phagosome pathway exhibited the highest degree of enrichment in the brain, followed by systemic lupus erythematosus, staphylococcus aureus infection, leishmaniasis, tuberculosis, and other pathways (Fig. 2d). The phagosome pathway is associated with lipid metabolism, signal transduction, and immune response. Antigen processing and presentation are related to lipid metabolism, redox homeostasis, and immune response. Based on GO analysis, The results showed that the membrane of cellular components (GO:0016020 and GO:0016021) exhibited the highest bar height, suggesting the most significant enrichment of this functional category in the gene set (Fig. 2e).

In order to further investigate core genes associated with improved behavior ability in ASO group, we conducted weighted correlation network analysis (WGCNA) on data from the brains of rats, consisting of 10,437 genes. This analysis identified 42 modules of regions and their respective hub regions (Fig. 3a, f). We tested the identified modules to assess their correlation with various traits, revealing that the MEgreen, MElightgreen and MEdarkgreen modules were significantly correlated with behavior (Fig. 3f). We applied the bottleneck algorithm using the plug of CytoHubba in Cytoscape software and found a total of 20 hub genes (Fig. 3b). Twelve genes were obtained by further expressed screening with $p < 0.05$, including *Trappc2, Spcs3, Naf1, Napg, Rab7a, Bark1, Atp6v1c1, Tnk2, Fgf13, Rexo2, Fabp3* and *Kansl2* (Fig. 3c). We constructed a graphical representation of the correlation network among 12 genes ($p < 0.05$). Positive correlations between genes were depicted in red, while negative correlations were indicated in blue (Fig. 3d). Furthermore, we observed the involvement of *Rexo2* and *Fabp3* in oxidative stress

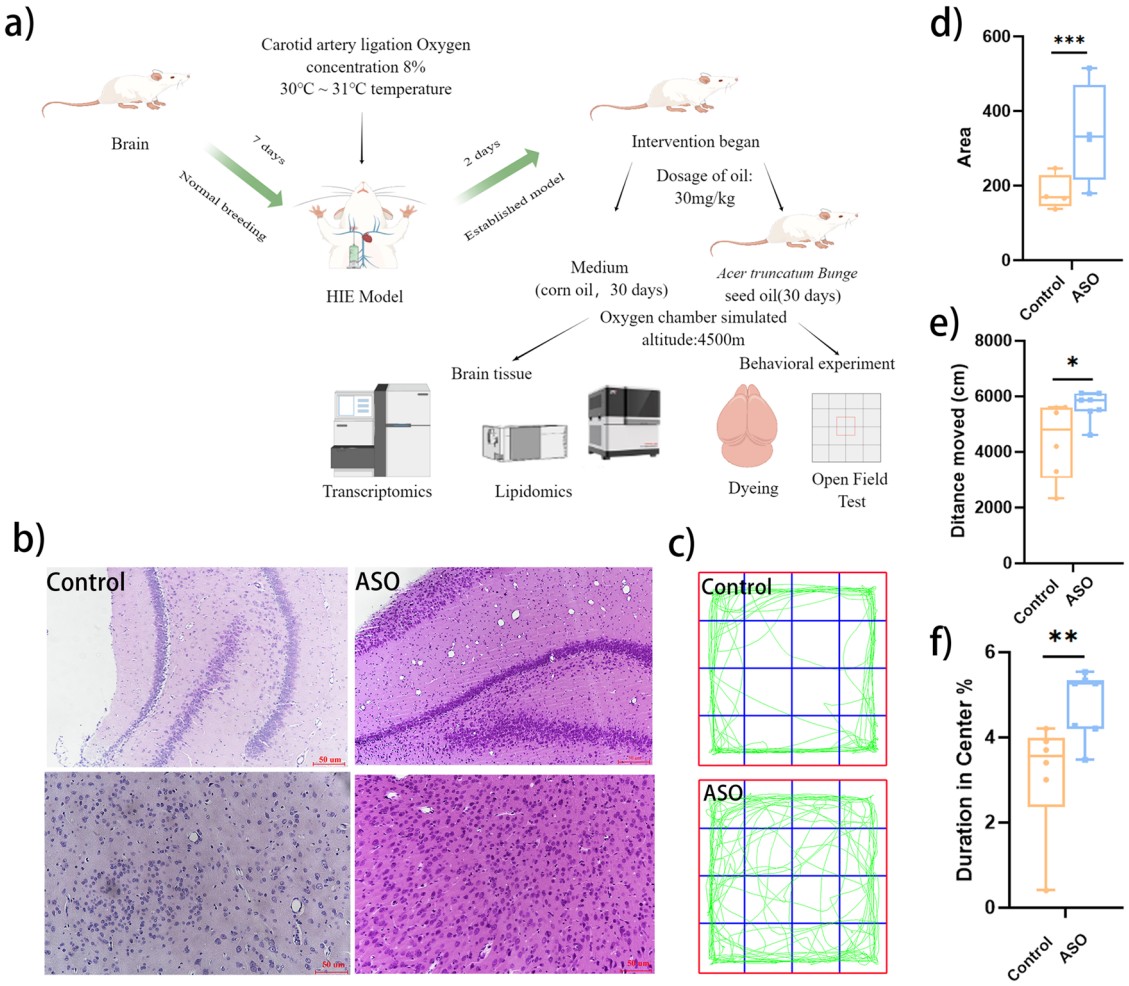

**Fig. 1 ASO ameliorated brain damage and improved behavior ability. a** Dosage regimen and sampling schedule. **b** HE staining showed histological changes in each group, and all three replicates had similar staining results. **c** The track of the open field experiment. **d** Quantitative analysis of HE staining. **e** The representative trajectory of the total distance. **f** Outcomes of the open field test, including distance moved and duration in center. *, **, *** indicates *p* value <0.05, <0.01, <0.001.

and lipid metabolism (Fig. 3e). Except for *Tnk2*, all gene expression levels in the ASO group were higher than those in the Control group among 12 genes (Fig. 3g–r).

**Free fatty acids profiling of rat brain**. We used a targeted lipid profiling approach to investigate changes in 32 free fatty acids (FFAs) in the rat's brain tissue, and 16 of them were significantly decreased in ASO group compared with Control group. There were significant differences between ASO and Control groups in ω-3, ω-6, and ω-9 FFAs (Table 1).

**Untargeted lipid profiling in brain between ASO and Control groups**. Brain samples from the Control group and ASO group rats were collected for analysis using an untargeted lipidomics approach. Using Progenesis QI 2.0, 13,400 lipid features were determined in the rat's brain and were obtained under rigorous quality control. In the positive ion mode, the corresponding distribution was 6216, while in the negative ion mode, 7184 were found. A total of 120 potential markers were identified according to the *p* < 0.05 and VIP > 1.

The FAs of structural lipids identified in 120 lipid metabolites showed significantly higher levels of structural ω-3 and ω-9 FAs in ASO compared to Control group (Table 2), and noteworthily ASO group exhibited decreased FFAs of ω-6 and ω-9 according to

results of FFAs targeted profiling (Table 1). There were differences in FAs unsaturation between two groups, in which ASO group has more polyunsaturated fatty acids (PUFA) and monounsaturated fatty acids (MUFA) than Control group (Table 2).

To discover different lipid molecules between Control and ASO groups, we conducted a multivariate statistical analysis using orthogonal partial least squares discriminant analysis (OPLS-DA) (Fig. 4a). S-Plots derived from the OPLS-DA model and volcano plots were employed to generate a list of features that were important for group discrimination (Fig. 4b, c). A total of 120 different metabolites were screened between the Control and ASO groups, which were classified by glycerophospholipids (GP), lysophospholipids, plasmalogen, and other lipids (Fig. 4d–g). Within the category of glycerophospholipids, the ASO group demonstrated a notable increase in the levels of phosphatidyl-cholines (PC), phosphatidylethanolamines (PE), phosphatidylser-ines (PS), and phosphatidylinositols (PI) compared to the Control group. Conversely, the levels of phosphatidyl glycerol (PG) and phosphatidic acid (PA) exhibited a decrease (Fig. 4d–j). The ASO group also exhibited increased lysophosphatidylcholine (LPC) and lysophosphatidylethanolamine (LPE), whereas decreased PS-O, PG-O and PI-O compared with Control group (Fig. 4f, g). The FA synthesis pathway displayed structural ω-3, ω-6, and ω-9 FAs were significantly increased after ASO administration, including

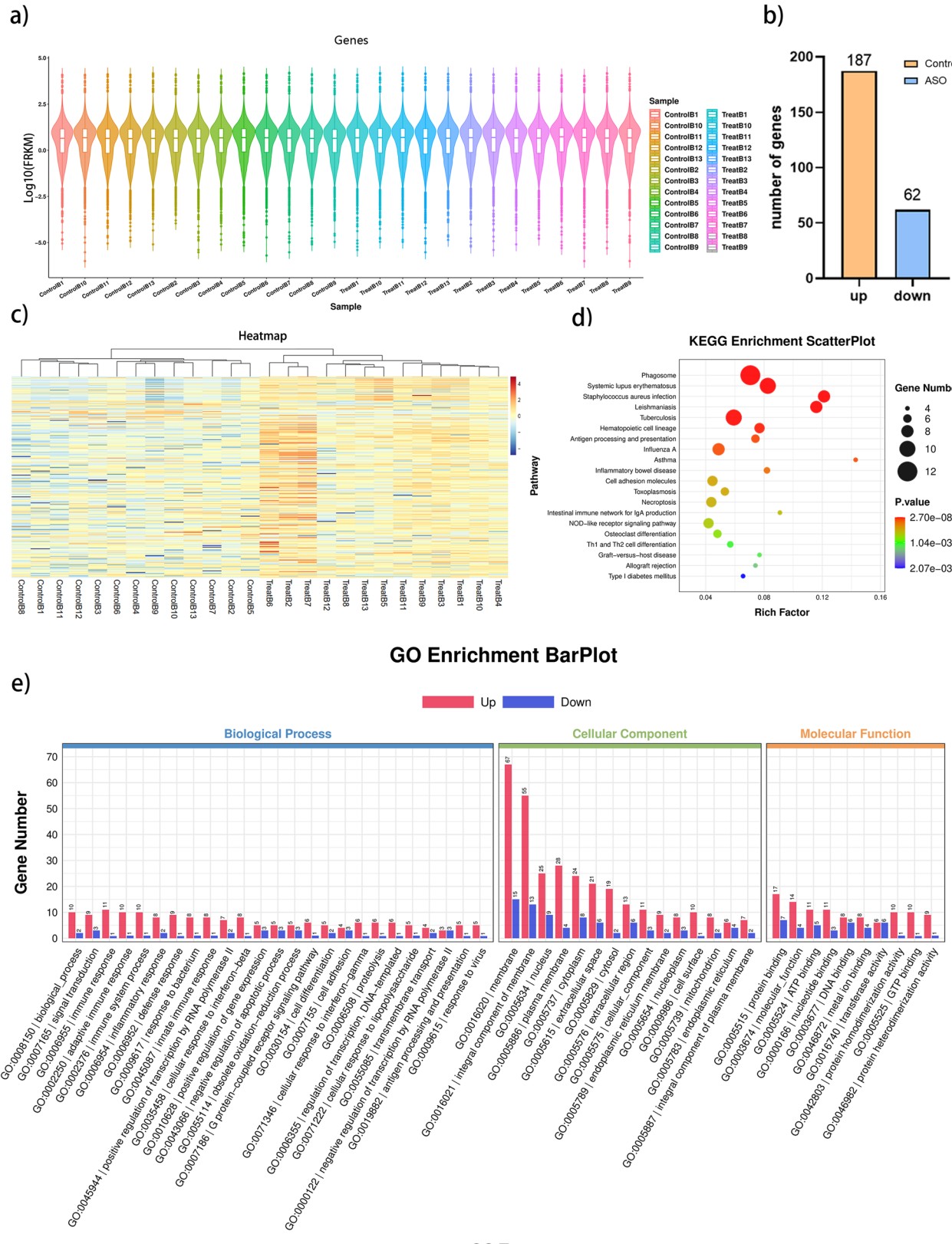

**Fig. 2 Transcriptome and DEG analysis of the rat's brain. a** Violin representation of gene expression levels. **b** Bar graph of the number of up-regulated and down-regulated genes. **c** Heatmap of differentially expressed genes. **d** The bubble chart of KEGG enrichment analysis. The color of the bubble represents the $p$ value (or $Q$-value) of the significance in enrichment. The smaller the $p$ value (or $Q$-value), the more significant the enrichment. **e** GO enrichment analyses of differential genes.

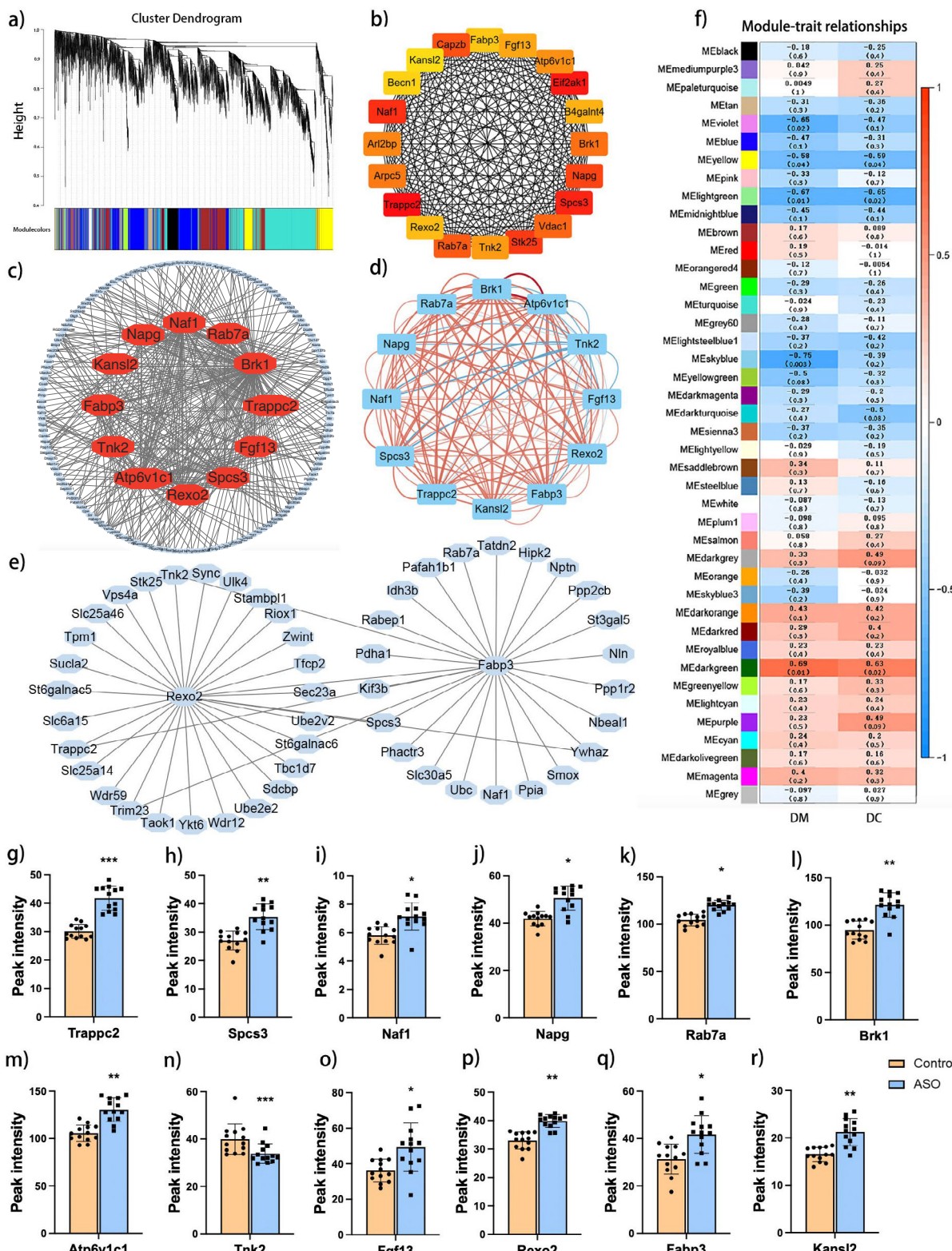

**Fig. 3 Weighted correlation network analysis (WGCNA) of rat's brains. a** Visualization of WGCNA. **b** The top 20 genes screened by CytoHubba in MCC. **c** Correlation analysis among the hub genes. **d** Diagram of the correlation network between 12 genes (*p* value < 0.05). **e** The gene associated with *Rexo2* and *Fabp3*. **f** Module identification by WGCNA. The heatmap colors are representative of the correlation between the module eigengenes and DM and DC. DM distance moved, DC duration in center. **g–r** The boxplots for core 12 gene expressions. *, **, *** indicates *p* value <0.05, <0.01, <0.001.

NA, docosapentaenoic acid (DPA), docosahexaenoic acid (DHA), but there was no difference in arachidonic acid (AA) (Fig. 4h). The well-defined clustering in the heatmap indicated the differential metabolites as biomarkers distinguished the two groups well (Fig. 4i). The present study elucidated the lipid metabolism transformation pathway in rats following ASO administration during HIE at high-altitude, revealing the active involvement of glycerol phospholipid and sphingolipid metabolism in this process (Fig. 4j).

**Combined pathway analysis with transcriptome and lipidome.** Seven genes were up-regulated in ASO group compared to Control group according to the criteria $p < 0.01$ and $Q < 0.01$, and also related to phospholipid metabolism and lipid peroxidation (Fig. 5a–g). The only one of the differential metabolites associated with COA was trans-2-Enoyl-OPC4-COA, which was higher in the ASO group than Control group. Also, there were significant differences in oxidized glycerol phospholipids (OxGPs), among which PI (27:2) + 2O and PI (30:3) + 3O were lower in ASO group than in Control group (Fig. 5h, i). In the depicted metabolic network diagram, differential metabolites are represented by circles and are located in the upper part of the figure. The genes displayed in the lower part of the diagram, connected by lines, were identified through a screening process and found to be associated with the metabolism of the differential metabolites (Fig. 5j). The circular plots of correlation between genes and metabolites showed that PI, PE, PC, and PS were significantly associated with differentially expressed genes. We investigated the correlation between three metabolites and genes, and the results showed that PI (27:2) + 2O and PI (30:3) + 3O were negatively correlated with *Dhfr, Abca12, Acsm5, Asf1b,* and *RGD1564801.* However, they were positively correlated with *Zbtb37.* Additionally, *Lpcat1* and *RGD1564801* showed a positive correlation with trans-2-Enoyl-OPC4-COA. *Zbtb37* exhibited a negative correlation with trans-2-Enoyl-OPC4-COA (Fig. 5k).

The integrated investigation involving lipid profilings and gene expressions conducted the mechanism of ASO affecting neuroprotection to phospholipid synthesis under the condition of HIE at high-altitude (Fig. 6a, blue layout) and FA metabolism (Fig. 6a, green layout), as well as lipid peroxidation (Fig. 6a, purple layout). The green layout was the process from FA to FA-CoA, the blue layout was the synthesis and metabolism of phospholipids, and the purple layout was the process of preventing lipid peroxidation (Fig. 6a). In the process from FAs to FA-CoA (green layout), up-regulation of the thromboxane A synthase 1 gene (*Tbxas*1, Fig. 6a) advanced the transformation of FAs to prostaglandin[20], while the increased expression of acyl-coenzyme A synthetase gene (*Acsm*, Fig. 6a) also promoted the transformation of FAs to FA-CoA[21], and further synthesis of structural FAs. Recombinant Lysophosphatidylcholine Acyltransferase 1 gene (*Lpcat1*, Fig. 6a) played a role in the synthesis of FA-COA into GPs-FA for synthesis of phospholipids, while Rattus norvegicus phospholipase A2, group XV (*Pla2g15*, Fig. 6a)

**Table 1 Targeted measurement of FFAs.**

| FFAs | Mean (Control) | Mean (ASO) | FC (ASO/Control) | p value |
|---|---|---|---|---|
| FA11:0 | 6.04 | 4.91 | 0.81 | 0.001*** |
| FA12:0 | 23.77 | 21.14 | 0.89 | 0.048* |
| FA14:1 | 10.35 | 8.17 | 0.79 | 0.001*** |
| FA14:0 | 51.95 | 41.51 | 0.80 | 0.003*** |
| FA15:0 | 25.67 | 19.14 | 0.75 | 0.000*** |
| FA16:0 | 952.42 | 764.15 | 0.80 | 0.001*** |
| FA16:1 | 79.78 | 65.09 | 0.82 | 0.001*** |
| FA17:0 | 25.87 | 20.45 | 0.79 | 0.003** |
| FA17:1 | 12.04 | 9.73 | 0.81 | 0.003** |
| FA18:0 | 671.34 | 544.29 | 0.81 | 0.001*** |
| FA18:1 | 623.29 | 500.42 | 0.80 | 0.003** |
| FA18:2 | 180.69 | 150.94 | 0.84 | 0.022* |
| FA18:3 | 3.69 | 3.36 | 0.91 | 0.192 |
| FA 20:0 | 0.98 | 0.93 | 0.95 | 0.401 |
| FA 20:1 | 10.30 | 10.55 | 1.02 | 0.389 |
| FA 20:2 | 30.18 | 26.40 | 0.87 | 0.036* |
| FA 20:3 | 54.50 | 47.96 | 0.88 | 0.076 |
| FA 20:4 | 588.41 | 478.61 | 0.81 | 0.002** |
| FA 20:5 | 1.56 | 1.72 | 1.10 | 0.203 |
| FA 21:0 | 7.62 | 7.58 | 1.00 | 0.456 |
| FA 22:0 | 0.09 | 0.11 | 1.21 | 0.100 |
| FA 22:1 | 0.42 | 0.53 | 1.26 | 0.119 |
| FA 22:2 | 0.25 | 0.23 | 0.94 | 0.270 |
| FA 22:3 | 4.66 | 4.36 | 0.94 | 0.252 |
| FA 22:4 | 177.30 | 144.89 | 0.82 | 0.014* |
| FA 22:5 | 26.10 | 19.23 | 0.74 | 0.009** |
| FA 22:6 | 215.88 | 186.01 | 0.86 | 0.063 |
| FA 23:0 | 0.11 | 0.14 | 1.32 | 0.077 |
| FA 24:0 | 0.14 | 0.18 | 1.31 | 0.090 |
| FA 24:1 | 0.04 | 0.05 | 1.20 | 0.179 |
| FA 25:0 | 0.03 | 0.04 | 1.26 | 0.070 |
| FA 25:1 | 0.11 | 0.15 | 1.36 | 0.058 |
| ω-3 Fatty acid | 3317.89 | 2696.69 | 0.81 | 0.001** |
| ω-6 Fatty acid | 3308.18 | 2680.48 | 0.81 | 0.001** |
| ω-9 Fatty acid | 2604.17 | 2111.60 | 0.81 | 0.001** |

FC means fold changes.
*, **, *** indicates p value <0.05, <0.01, <0.001.

**Table 2 Total structural fatty acids composition of brain tissue (Control vs. ASO).**

| FAs | Mean (Control) | Mean (ASO) | Fold (ASO/Control) | p value |
|---|---|---|---|---|
| ω-3 Fatty acid | 2,241,105.52 | 2,393,245.99 | 1.07 | 0.003** |
| ω-6 Fatty acid | 778,559.16 | 781,879.51 | 1.00 | 0.420 |
| ω-9 Fatty acid | 494,188.03 | 533,527.13 | 1.08 | 0.002** |
| MUFA | 375,279.76 | 407,031.16 | 1.08 | 0.006** |
| PUFA | 811,727.87 | 857,730.33 | 1.05 | 0.007** |
| UFA | 1,181,577.76 | 1,258,241.38 | 1.06 | 0.0009*** |
| SFA | 1,312,430.39 | 1,355,716.24 | 1.03 | 0.0717 |
| UFA/SFA | 0.90 | 0.93 | 1.03 | 0.126 |
| MUFA/SFA | 0.28 | 0.3055841 | 1.05 | 0.061 |
| PUFA/SFA | 0.61 | 0.63 | 1.02 | 0.219 |
| MUFA/PUFA | 0.46 | 0.47 | 1.03 | 0.86 |

**, *** indicates p value <0.05, <0.01, <0.001.

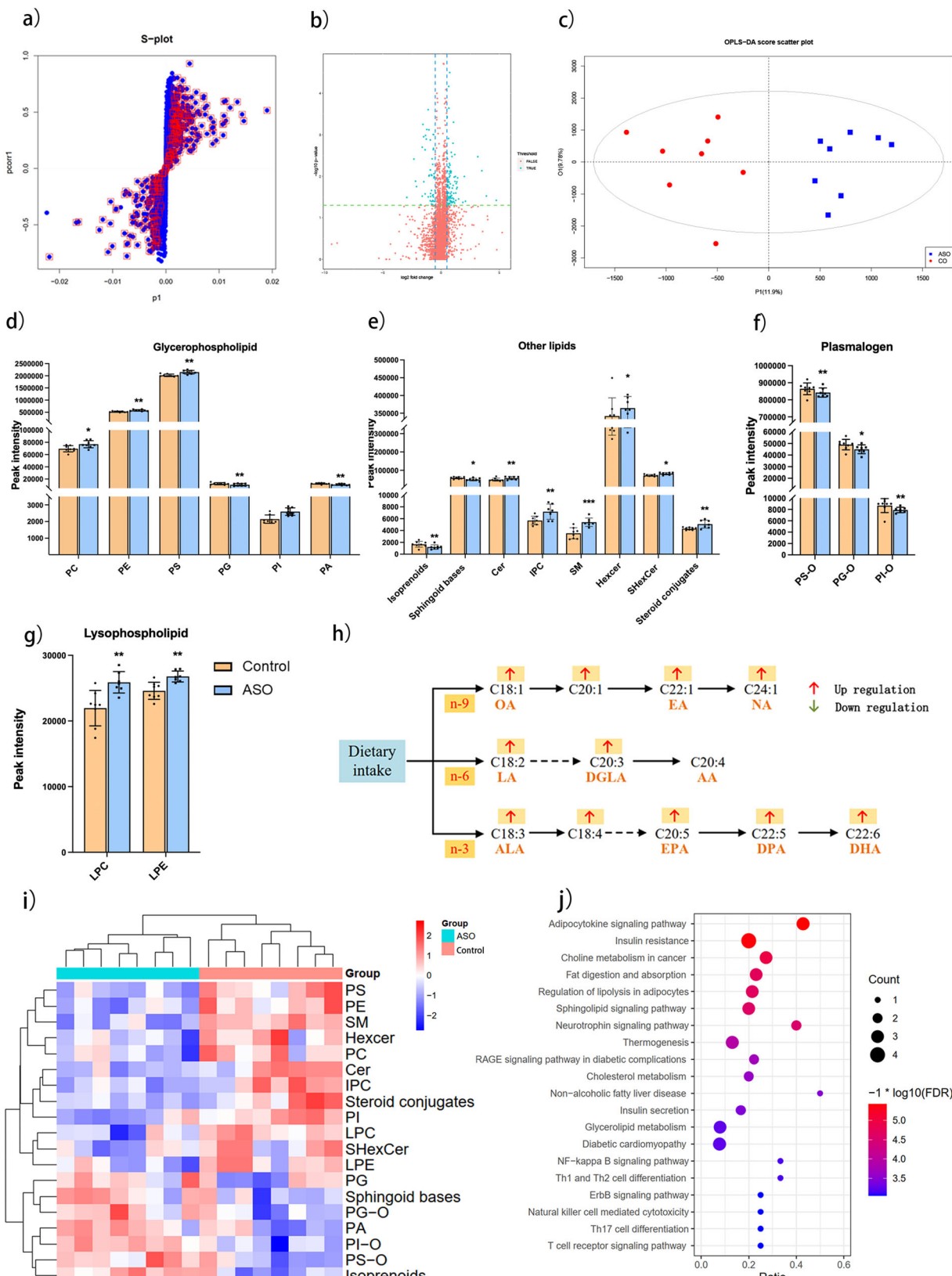

**Fig. 4 Alterations in the metabolism induced by ASO treatment. a** Score plot obtained by OPLS-DA. **b** Volcano plot of metabolites. **c** OPLS-DA score scatter plot of brain lipidomics. Lipid classification. **d** Glycerophospholipid. **e** Other lipids. **f** Plasmalogen. **g** Lysophospholipid. **h** Changes of structural fatty acid chains after ASO administration. **i** Heatmap of the differential lipid headgroups. **j** The bubble chart of KEGG enrichment. *, **, *** indicates *p* value <0.05, <0.01, <0.001.

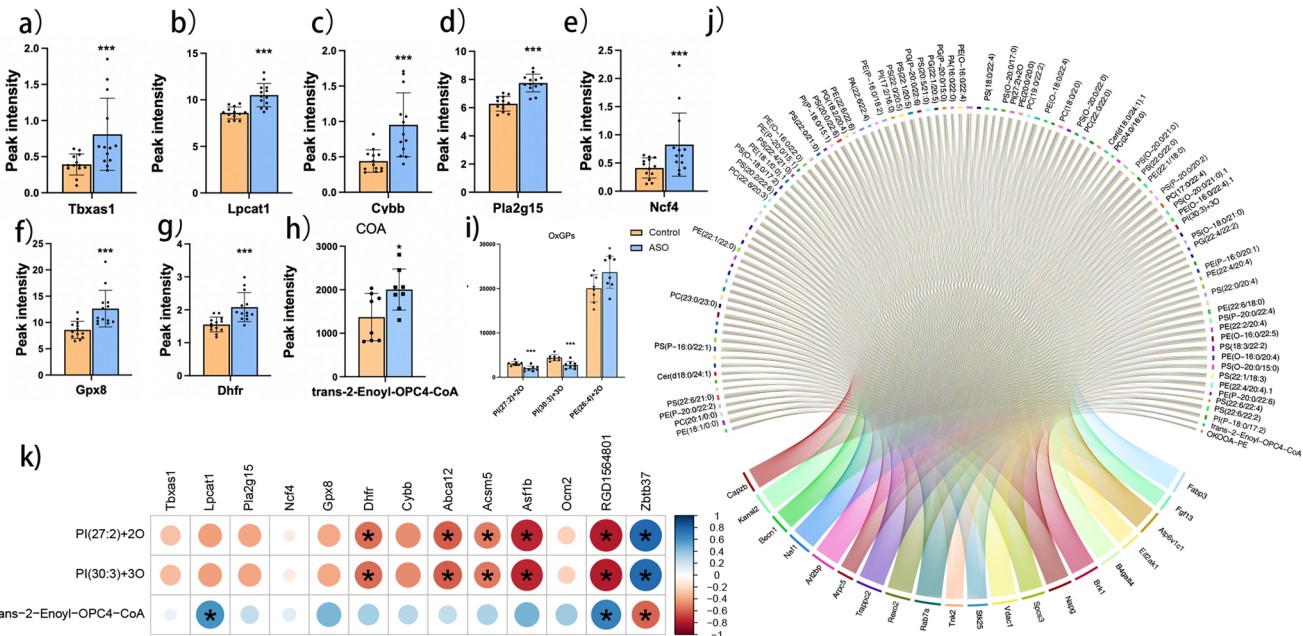

**Fig. 5 Network diagram combining transcriptomics with lipidomics analysis. a–g** Potential functional genes. **h, i** Candidate metabolite biomarkers. **j** Circle plots of the gene and metabolite correlations. **k** Correlation diagram of candidate genes and metabolites.

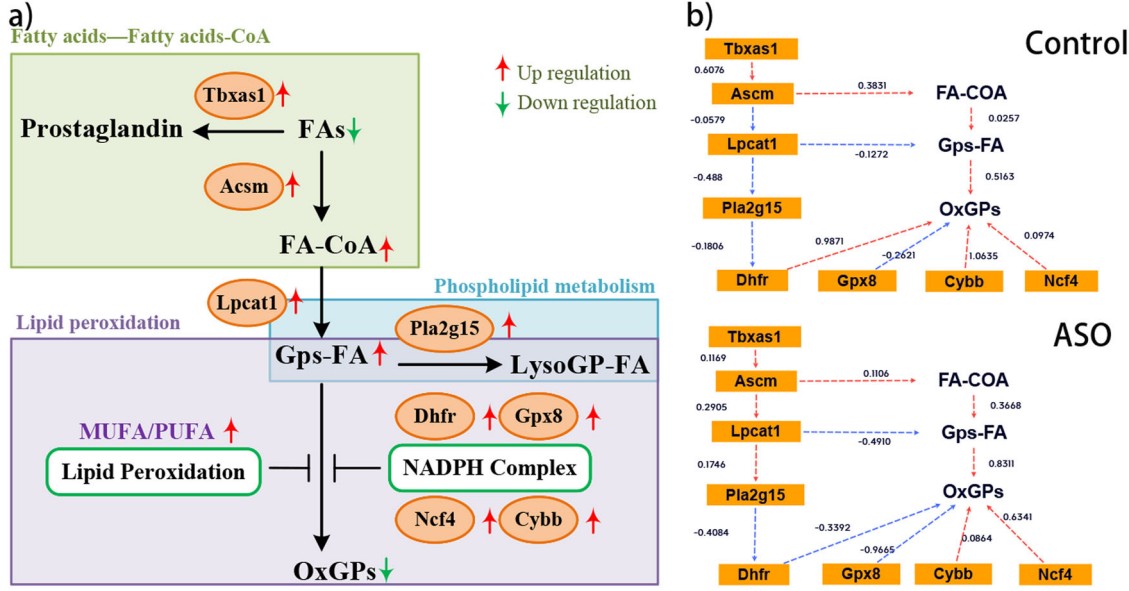

**Fig. 6 Diagram of the mechanism of the possible effects of ASO intervention on neuroprotection. a** Possible mechanisms of action of ASO. **b** Customizing structural equation modeling plots.

participated in the metabolism of phospholipids. Cellular GPs-FAs were easily converted to OxGPs under stress, indicating the severity of oxidative stress[22]. Dihydrofolate reductase (*Dhfr*) glutathione peroxidase 8 (*Gpx8*), neutrophil cytosolic factor 4 (*Ncf4*), and Cytochrome b-245 beta chain (*Cybb*) were all genes that synthesize the nicotinamide adenine dinucleotide phosphate (NADPH) complex, and their rise may result of an increase in NADPH[23–25]. Besides, MUFA can inhibit PUFA oxidation well[26], which also participated in ameliorating the excessive lipid peroxidation (Table 2 and Fig. 6a). The structural equation modeling results revealed that the ASO group exhibited an increase in the pathway from FA-CoA to Gps-FA. Moreover, ASO enhanced the association between *Ascm-Lpcat1-Pla2g15* and weakened the association between *Dhfr*, *Gpx8* and *Cybb* to

OxGPs, which was consistent with the pathway results. It was also noted that the role of *Dhfr* and *Gpx8* in the OxGPs pathway went from being promoting factors to inhibitory factors (Fig. 6b).

## Discussion

Hypoxic-ischemic encephalopathy (HIE) is a leading cause of neonatal death and neurological impairment, particularly in high-altitude regions, where effective treatments remain elusive. Although ASO has been observed to improve brain function and enhance memory, learning, and cognition, its underlying cellular and molecular mechanisms remain unclear. The integration of transcriptomics and lipidomics provides an unprecedented opportunity to investigate the multi-targeted effects within the body. This approach allows connecting tissue pathology with

alterations in gene expression, offering valuable mechanistic insights. In this study, we reported the protective effects of ASO on neonatal rats with HIE in a plateau environment and demonstrated the preliminary molecular mechanism by integrating transcriptomics and lipidomics strategies. The core functional genes were identified (Fig. 3g–r), and targeted lipid profiling indicate a decrease in FFAs of ω-6 and ω-9 (Table 2), while untargeted lipid profiling revealed differential metabolites associated with fatty acid metabolism and phospholipid synthesis (Fig. 4). Furthermore, we mapped out the pathways related to fatty acid synthesis, phospholipid metabolism, and lipid peroxidation (Fig. 6).

FFAs serve as biomarkers of numerous brain disorders and are also associated with cognitive performance[27]. Previous research has indicated that elevated levels of free NA are coupling with leukoencephalopathy and mental illness[27]. Patients exposed to high-altitudes have exhibited microbleeds in the corpus callosum[28,29]. And the accumulation of FFAs in the rat's brain is considered an indicator of small blood vessel rupture[27]. As a result, oral medications containing free NA leading to FFAs accumulation may increase the above risks[30]. However, administration of ASO including lipid structural NA contributes to improve cognitive ability and rebuild myelin sheath[8]. In our study, supplementation with ASO was found to significantly reduce free fatty acid levels (Table 1), and enhance exercise capacity and reduce brain damage proved by HE staining and behavioral testing (Fig. 1). Meanwhile, FFAs, especially ω-6 FAs, are thought to be a significant role in cellular oxidative damage resulting from lipid peroxidation, and reducing the production of free AA is considered advantageous in reduce cellular oxidative damage (Table 1), which is consistent with Moustafa's study[31].

Many studies, including our previous research, have indicated that ASO contributes to brain development[8,14,32]. However, due to lipidic nature of ASO, its composition is intricate, rendering the investigation of its mechanism challenging[8]. The utilization of multi-omics approaches for investigating complex systems is predominantly observed in pharmacological studies of herbal medicines and even plant extracts[18,19]. According to Laufer's study, we employed WGCNA for module association analysis, enabling the rapid identification of hub genes, even when dealing with high throughput data[33]. Then lipidomics combined with transcriptomics strategies were carried out for the first time to elucidate the initial mechanism of ASO intervention in plateau HIE rats (Fig. 6). From FA to GPs-AA, through Acsm (Fig. 5k), Lpcat1 (Fig. 5b) and trans-2-Enoyl-OPC4-C (Fig. 5h), the changes of gene expressions and metabolites displayed obvious and consistent trends. Our identification of the synthesis pathway, from FAs to GPs, likely represents the workflow by which ASO carries out in vivo functions (Fig. 6). This mechanism notably indicates ASO administration enhances phospholipid and essential structural fatty acid synthesis, thereby contributing to the enhancement of autonomic motor abilities[34].

Fatty acid binding proteins (Fabps), with a molecular weight of 14–15 kDa have an affinity for hydrophobic ligands, particularly long-chain fatty acids[35]. In addition to regulating lipid solubility, mobility, and utilization, Fabp3 is also thought to function as a lipid "chaperone"[36]. Previous research has demonstrated that a partial loss of Fabp3 in mice leads to a reduction in fatty acid uptake[37]. In our study, we observed higher Fabp3 levels in the ASO group compared to the Control group, aligning with previous findings[37], indicating that the administration of ASO may improve the in vivo utilization rate of FAs. Furthermore, our results revealed elevated expression of Acsm5 in the ASO group compared to the Control group, which catalyzes the FAs by CoA to produce acyl-CoA as the first step in FAs metabolism[38] (Supplementary Fig. 2e). The reduction in Zbtb37 levels

contributes to enhanced overall cellular transcriptional activity, consequently fostering the growth and development of the neonatal rat[39,40] (Supplementary Fig. 2f). The ABCA subfamily is recognized for its role in lipid transport. Abca12, as a transmembrane lipid transporter gene in keratinocytes, participates in lipid transport through lamellar granules[41] (Supplementary Fig. 2g). Additionally, Ocm2 is associated with the growth and development of neurosynapses[42] (Supplementary Fig. 2h). Abca12 and Ocm2 genes expression showed an increase in ASO group compared to the Control group, suggesting that ASO supplementation may be associated with lipid transport and neurodevelopment.

Due to the diminished atmospheric pressure at high altitudes, cells, tissues, and organs, receive a reduced supply of oxygen, leading to a decrease in the oxygen saturation of arterial blood[43]. To improve cognitive performance, reducing oxidative stress is considered a crucial mechanism for neurological repair at high altitudes, especially in cases of HIE[44–47]. Additionally, studies have displayed that Ppp1r may contribute to the generation of reactive oxygen species and subsequent lung damage caused by acrolein[48]. The Rex acts as a redox sensor in response to intracellular NADH/NAD+ [49]. The Rexo2 gene regulates the cellular oxidative environment in response to NADPH/NAD+ [50,51]. Altitude induces a spectrum of oxidative stress markers, including Ppp1r3e, Ppp1r4a, Ppp1r16b, etc[52,53]. Our findings showed that ASO administration resulted in inhibition of lipid oxidation, evidenced by down-regulated expression of antioxidant genes in Rexo2, Ppp1r3e, Ppp1r4a, Ppp1r16b, coupling with decreased levels of OxGPs (Supplementary Fig. 2b–d). All these results suggest that ASO intake significantly reduces cellular oxidative stress in HIE rats at high altitudes.

Hypoxia-induced oxidative stress enhances vascular endothelial permeability, promotes leukocyte adhesion, and is accompanied by alterations in endothelial signaling pathways and the redox-regulated transcription factor Nfkb[54]. Nfkb is associated with high-altitude cerebral and pulmonary edema. Quercetin, a plant extract, has been reported to reduce cellular oxidative damage and alleviate hypoxic injury at altitude by inhibiting Nfkbia[54,55]. Interestingly, we observed a notable decrease in the expression of Nfkbia following ASO administration (Supplementary Fig. 2a). ASO intake induced high concentrations of long-chain MUFA, including NA (Table 2). The consumption of NA is advantageous for white matter repair and brain development[56,57]. In the high-altitude hypoxic environment, the incidence of white matter lesions is relatively elevated[58–60]. It is hypothesized that ASO may enhance the autonomic motor of HIE rats in the high-altitude hypoxic environment, possibly due to white matter repairing. Further investigation in the field of cytobiology is planned to explore this hypothesis in the future.

It is worth noting that the TTC staining plot showed that unilateral carotid artery ligation led to bilateral cerebral infarction (Supplementary Fig. 1a), which has been reported in the literature as a possible cause of impaired lateral branch circulation[61,62]. In addition, in our previous study, we reported the results of HE staining and metabolic analysis in sham surgery rats, which also confirm the success of HIE model[63]. Due to the limited availability of samples, we are unable to perform comprehensive validation procedures for numerous biomarkers, including genes expression and lipids calculation, after transcriptomic and lipidomic assays. Future studies will focus on the mechanism of ASO-mediated lipid peroxidation in HIE, and the genes, proteins and metabolites of our target pathway will be validated. In summary, our study demonstrated preliminary metabolism mechanism of ASO ingesting in rats, and the potential to develop supplement of ASO for HIE neonates under high-altitude.

## Materials and methods

**Experimental animals and treatments.** All animal experiments were performed according to protocols approved by the Scientific Research Ethics Committee of Shengjing Hospital affiliated with China Medical University. Sprague-Dawley pregnant rats at gestation days 18–21 were purchased from Liaoning Changsheng Biotechnology Co. Ltd (Permit number: SCXK Liaoning 2020-0002). All rats were housed in a temperature-controlled room (22–26 °C) under 12 h light and dark cycles, with free food and water throughout the study. All experiments performed in this study were following the Guide for the Care and Use of Laboratory Animals and were approved by the Ethics Committee of Medical ethics committee of Shengjing Hospital of China Medical University (No. P2020PS661K).

To investigate the neuroprotective effects of ASO on HIE, the rats were exposed to simulated 4500-meter altitude. After successful modeling for 48 h, the HIE rats were reclassified into two groups: Control group ($n = 35$) and ASO group ($n = 35$). HIE rats in the ASO group underwent intragastric ASO treatment, whereas HIE rats in the Control group received an equivalent volume of corn oil as the control vehicle[64]. After a 30-day intervention period, eight rats from each group were subsequently chosen randomly for behavioral evaluation. Unfortunately, during the experiment, two rats in the control group and one rat in the ASO group died. In addition, we collected separately all remaining rat brain tissue from each of the two groups for histological evaluation ($n = 6$), lipidomics ($n = 8$), and transcriptomic studies ($n = 13$). Especially, before the ASO intervention, we conducted a detailed examination and comparison of the Sham group ($n = 8$) and the Model group ($n = 8$), which was consistent with our previous study[63]. All samples were stored at −80 °C before use. Based on the recommended daily intake of NA is 0.3 g/day, ASO and vehicle (corn oil) were administered to rats at 30 mg/kg by gavage, according to the equivalent dose calculated on the body weight of humans and animals. ASO was extracted from *Acer truncatum Bunge* seed extract by supercritical[65] and provided by Baofeng Biotechnology (Beijing) Co., Ltd., and the component test report was issued[63].

**Rats model preparation of HIE.** The animal model of HIE was performed according to the Levine-Rice method[66]. All postnatal 7-day-old rats of both genders were anesthetized with ether inhalation. An incision was made in the midline of the neck, and the left common carotid artery was exposed, isolated from the nerve and vein, and ligated with 6/0 surgical silk. The artery was then cut between the ligations. Sutures were then used to close the wound, and the animal was allowed to recover. Each surgery was controlled within 5 min, and the body temperature of the animal was maintained at 37 °C using a temperature blanket during the operation. After 2 h of recovery with their mothers in cages, the pups were placed in a lower oxygen chamber (6% oxygen balanced with 94% nitrogen) for 2.5 h at 35–37 °C.

**Pathophysiological test**

*TTC.* The rats were sacrificed by cervical dislocation, and their brains were directly removed. After being frozen in a refrigerator at −20 °C for 5 min, the brains were sliced into serial sections with a thickness of 2 mm. The first incision was made at the midpoint of the connection between the anterior pole of the brain and the optic chiasm. The second cut was at the optic chiasm, the third cut was in the funnel handle, and the fourth cut was between the tail pole of the posterior lobe and the funnel stalk, cutting five or six pieces. The sections were placed in 1% TTC, shielded from light, and placed in a 37 °C temperature chamber for 15–30 min. The sections were uniformly stained under light

and then fixed with 4% paraformaldehyde. The brain slices-stained red represented normal brain tissue, while the white area indicated infarcted brain tissue.

*HE staining.* HE staining was performed according to standard procedures[67]. Following that, 5-μm sections were cut and stained with hematoxylin and eosin. The mounted slides were viewed and photographed using an Olympus BX53 fluorescent microscope (Tokyo, Japan). The staining intensity of the trabecular bone was assessed Image-Pro Plus 6.0 software and reported as an IOD value. The results of three repeated stained samples were selected for statistical analysis, and four domains ($1002 \times 1008$ pixels) were selected for each sample for quantitative analysis.

*Open field test (OFT).* Rats were placed in the open field reaction box and allowed to engage in free activities. The OFT was an experiment that was used to assess general locomotor activity levels and anxiety in rodents. Each rat was placed in the center of the open field apparatus and moved freely for 5 min. The average speed and time/distance in the center were recorded to measure the locomotor activity and anxiety levels. Between each trial, the maze was wiped clean with a damp sponge and dried with paper towels.

*Transcriptomic analysis.* A total cellular RNA was extracted from rat's brain tissue and used for $2 \times 150$ bp paired-end sequencing on an Illumina Novaseq 6000 (LC-Bio Technology CO., Ltd) in accordance with the vendor's recommendations. Aligning RNA-sequencing reads to the murine reference genome (https://ftp.ensembl.org/pub/release-101/fasta/-mus musculus/dna/) was performed using HISAT2 (https://daehwankimlab.github.io/hisat2/,version:hisat2-2.0.4). The mapped reads were assembled using StringTie (https://ccb.jhu.edu/software/stringtie/,version:stringtie-1.3.4). In the following step, all transcriptomes were merged using gffcompare software (https://ccb.jhu.edu/software/stringtie/gffcompare.shtml,version:gffcompare-0.9.8.) to reconstruct a comprehensive transcriptome. Following the generation of the transcriptome, StringTie and Ballgown (https://www.bioconductor.org/packages/release/bioc/html/ballgown.html) were used to estimate the expression levels of all transcripts as FPKM (FPKM = (total exon fragments/mapped reads (millions) × exon_length(kB))). Differentially expressed genes (DEGs) were identified by at least 2 fold changes and $Q$-value < 0.05 (DESeq2 R package, https://www.bioconductor.org/packages/release/bioc/html/-DESeq2.html). An analysis of gene differential expression was performed using DESeq2 software between two groups (and by edgeR between two samples). The genes with the parameter of false discovery rate (FDR) < 0.05 and absolute fold change ≥2 were considered differentially expressed genes. Genes exhibiting differential expression were then analyzed for GO enrichment.

**Lipidomics analysis**

*Targeted lipidomics.* Targeted lipidomics analysis were performed using standard operating procedures derived from previously published methods[68–71]. Samples were analyzed by ESI using ACQUITY UPLC (Waters) and Xevo TQ-S (Waters, Milford, MA, USA) mass spectrometry (Waters). Lipid separation was performed on a Waters UPLC BEH C8 column ($2.1 \times 100$ mm, 1.7 μm, Waters) with a gradient mobile phase consisting of phase A (acetonitrile: water = 1:10, 0.1% acetic acid, 1 mM ammonium acetate) and phase B (isopropanol: acetonitrile = 1:1). A 20-min accelerated elution curve was used with a mobile phase flow rate of 0.26 ml/min. The 1 μl sample was eluted first with 90% A and 10% B, linearly eluted to 65% A and 35% B within 2 min, and

increased to 15% A and 85% B within 4 min. The gradient was further increased to 0% A and 100% B over the next 9.9 min, and then to 90% A and 10% B over 0.1 min. In the last part of the gradient, the amount was maintained at 90% A and 10% B. Mass spectrometry was performed using a Waters Xevo TQ-S (Waters, Milford, MA, USA) mass spectrometer. When the acquisition mode was positive ion (ESI+), the ion source voltage was 3.0 kV and the temperature was 150 °C. When the acquisition mode was negative ion (ESI−), the ion source voltage was −1.8 kV and the temperature was 150 °C. Desolvation temperature 350 °C, desolvation gas flow rate 1000 l/h. The voltage of the cone hole was 29.0 V and the gas flow rate was 150 l/h. The peak area of the targeting data was calculated using target Lynx quantitative software, and the retention time allowed an error of 15 s. Concentrations were calculated using the single-point internal standard method to obtain quantitative results.

*Untargeted lipidomics.* The samples were analyzed by ACQUITY UPLC (Waters) and XEVO-G2XS quadrupole time-of-flight (QTOF) mass spectrometry (Waters) with ESI. Lipid separation was performed on an Acquity UPLC charged surface hybrid C18 column (2.1 × 100 mm, 1.7 μm, Waters), and the gradient mobile phase was composed of 10 mM ammonium formate and 0.1% formic acid acetonitrile/aqueous solution (A, 60:40, v/v) and 10 mM ammonium formate and 0.1% formic acid isopropanol/acetonitrile solution (B, 90:10, v/v). A 20-min accelerated elution curve, with the flow rate of the mobile phase 0.4 ml/min, was employed in this study. The injected 1 μl sample was initially eluted with 40% B, graded linearly to 43% B in 2 min, and then increased to 50% B in 0.1 min. In the next 9.9 min, the gradient was further increased to 54% B and then increased to 70% in 0.1 min. In the last part of the gradient, the amount of B was increased to 99% in 5.9 min. Finally, solution B returned to 40% in 0.1 min, and the column was balanced for 1.9 min before the next injection. The lipids in both positive and negative modes were detected by a Xevo-G2XS QTOF mass spectrometer, which was operated in MSE mode from *m/z* 50–1200, and the collection time for each scan was 0.2 s. The source temperature was set to 120 °C. The desolvation temperature was 550 °C, the gas flow rate was 1000 l/h, and nitrogen was used as the flowing gas. The capillary voltage was 2.0 kV (+)/1.5 kV (−), and the cone voltage was 20 V. Leucine encephalin (molecular weight = 555.62 × 200 pg/μl, 1:1 acetonitrile: water) was used as the locking mass for accurate mass determination and corrected with 0.5 mm sodium formate solution. The samples were randomly sorted, and 5 quality control samples were initially injected to adjust the conditions of the column. A QC sample was injected in every 10 samples for analysis to investigate the repeatability of the data.

*Statistics and reproducibility.* Metabolic changes in plasma extract were analyzed by using the UPLC-Q-TOF MS system and the equipped software Progenesis QI (Waters). The original data were preprocessed, and the linear model was adjusted. Orthogonal Partial least squares discriminant analysis (OPLS-DA) was first used for classification discrimination. The reliability of the model was verified by cross-validation and displacement test. The parameters R2 and Q2 were used to evaluate the interpretability and predictability of the model, respectively. By *p* value ($p < 0.05$), variable importance projection (VIP > 1), and false discovery rate (FDR < 0.05), the standard potential difference marker is selected. The best truncation value was determined by using the youden index. Finally, potential biomarkers were correlated with metabolic pathways through KEGG. Customizing structural equation modeling plots was carried out by using the "semptools" package. All statistical analyses were performed using R version 3.6.3, and $p < 0.05$ was considered statistically significant.

**Reporting summary**. Further information on research design is available in the Nature Portfolio Reporting Summary linked to this article.

## Data availability

The corresponding author has full access to all the data in the study and take responsibility for the integrity of the data and the accuracy of the data analysis. The transcriptome data are stored in the SRA database, accession to cite for these SRA data: PRJNA909348. Temporary Submission ID: SUB12485094. SRA records will be accessible with the following link after the indicated release date: https://www.ncbi.nlm.nih.gov/sra/PRJNA909348. The relevant metabolomics raw data and master alignment ion feature tables generated for this study have been deposited in National Metabolomics Data Repository (NMDR) using metabolomics workbench (datatrack_id: 3673 study_id: ST002483, https://www.metabolomicsworkbench.org), where it has been assigned Study ID ST002483. The data can be accessed directly via its Project https://doi.org/10.21228/M87H9N. All source data for the figures and tables of this work are available online at Zenodo (https://doi.org/10.5281/zenodo.8155103)[72]. Requests for additional information or data can be addressed to the corresponding author.

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

## Acknowledgements

This study was funded by Shengjing Hospital of China Medical University through project NO. 112-3110119086. We especially thank the instrumentation support from the Department of Pediatrics, Shengjing Hospital of China Medical University.

## Author contributions

X.C., W.S. and J.L. conceptualized and designed the study. W.S., J.H., Y.G.S. and H.C. conducted statistical analyses, compound identification, and data visualization. S.L., X.X. and Y.F. raised animals and harvested samples. J.H., C.D., F.L. and Y.S. performed the data curation. W.S., J.H., Y.G.S. and H.C. performed investigation. X.C. and W.S. performed the supervision. J.H., W.S. and Y.G.S. performed the visualization and statistical analysis work. X.C. drafted the manuscript; J.H., W.S. and Y.G.S. revised the manuscript. All authors read and approved the final manuscript.

## Competing interests

The authors declare no competing interests.
