## [Peer Review File · Communications Biology]

Reviewers' comments:

Reviewer #1 (Remarks to the Author):

The manuscript entitled "Integrated lipidomics and transcriptomics reveal the neuroprotective effect of Acer truncatum Bunge Seed extract on hypoxic-ischemia encephalopathy rats exposed at high-altitude" is potentially interesting, however, there were plenty of descriptive mistakes and technical errors in this paper.

1. Moderate hypothermia is a clinically recommended and well established intervention for neonatal HIE, the authors placed the pups in a lower oxygen chamber at 30-33°C? We deeply doubt the neonatal HIE model can not be made successfully in this condition, and there was almost no infarction in model group as shown in Supplementary Figure 1 a. Furthermore, why did the both sides of brain have the infarction?
2. The authors stated "In this study, we found ASO administration rats group exhibited reduced cerebral edema compared to the Control group", but the cerebral edema in the Model group is not obvious.
3. In line 276-279, the authors should give more details about the method for HE staining. How many objectives did authors selected? How many fields were quantitated? How was the statistical significance achieved?
4. The authors stated "The results of the open field test displayed that the performance of distance moved and duration in center in ASO group was significantly more effective than the Control group, suggesting that the autonomic motor ability of mice was significantly improved after taking ASO" but there was a deduction in distance moved and duration in center in ASO group.
5. Supplementary figures 1a, 1c, 1d, and figure 3d were not described in results section.
6. The descriptions of figures 2d, 2e and 4j in the text is not clear.
7. In line 126-129, the authors stated phosphatidylglycerols (PG) and phosphatidic acids (PA) significantly increased in the ASO group, which is not consistent with figure 4d.
8. The description in line 146-149 is not consistent with figure 5k.

Reviewer #2 (Remarks to the Author):

This article is very interesting, but there are some issues that need to be addressed:

1. Figure 1: The colors in the HE-stained images in Figure 1 and Supplementary Figure 1 are inconsistent. Please check. Additionally, for the quantitative bar graphs, it is recommended to use scatter plots with raw data.
2. In Supplementary Figure 1, for the TTC-stained images, the Control group underwent HI modeling, so why does the image not show infarction? Also, quantitative analysis of TTC staining is needed. Additionally, why were only four groups analyzed for TTC, while other analyses were done for only two groups?
3. Figures 2-6: The image in Supplementary Figure 2 is not clear, and the text in the image is blurry.
4. Line 73: "Based on GO analysis, membrane of cellular component (GO:0016020 and GO:0016021) was activated in molecular function(Fig. 2e)." This description does not match the displayed image. Please verify.
5. Figure 2 and Figure 6 does not have asterisks (*) indicating statistical significance, so the explanations (*, **, ***) in the legend are not needed.
6. Line 127: "glycerophospholipids (GP), lysophospholipids, plasmalogen, and other lipids (Fig. 4D-G)." The format for Figure 4D-G is inconsistent. Besides, Line 220: (Supplementary Figure 2H) shows inconsistent expression.
7. Line 146: "We investigated the correlation between three metabolites and genes, and the results showed that PI (27:2)+2O and PI (30:3)+3O were positively correlated with Taappc2, Spcs3, Naf1, Napg, Rab7a, Brk1, Atp6v1c1, Rexo2, Fabp3, and Kansl2, and negatively correlated with Dhfr. There was a positive correlation between Lpcat1 and trans-2-Enoyl149 OPC4-COA, but a negative correlation between Trappc2 and trans-2-Enoyl-OPC4-COA (Fig. 5k)." Please check the description, as it does not

match the displayed image.

8. The Discussion section should avoid excessive repetition of the results.

9. In the Methods section, please provide the total number of animals used, as well as the number in each group, including sham and model groups.

10. In the Transcriptomic analysis and Lipidomics analysis, please provide the number of samples tested in each group and specify the specific brain regions. Additionally, it would be beneficial to include validation of key genes or lipids if possible.

Finally, I suggest that the authors make major revisions to this manuscript.

Response to Referees

Reviewer #1 (Remarks to the Author):

The manuscript entitled “Integrated lipidomics and transcriptomics reveal the neuroprotective effect of *Acer truncatum* Bunge Seed extract on hypoxic-ischemia encephalopathy rats exposed at high-altitude” is potentially interesting, however, there were plenty of descriptive mistakes and technical errors in this paper.

1. Moderate hypothermia is a clinically recommended and well established intervention for neonatal HIE, the authors placed the pups in a lower oxygen chamber at 30-33°C? We deeply doubt the neonatal HIE model can not be made successfully in this condition, and there was almost no infarction in model group as shown in Supplementary Figure 1 a. Furthermore, why did the both sides of brain have the infarction?

Dear sir or madam,

Thank you. This was a writing error and we have corrected it in the "Materials and methods" section of the new manuscript for the Rats model preparation of HIE. Since this project is a continuation of our previous work and the animal modeling method is the same as before, we have revised the temperature range from 30-33°C to 35-37°C¹ (highlighted in red at line 280).

We incorporated previous research findings and successfully developed a well-established modeling method for the rat model of HIE in our study²⁻⁴. Compared to normal controls, we have marked the damages in the TTC stain plot in supplementary Figure 1a, although it is not obvious. In our previous parallel study, hematoxylin and eosin (HE) staining were used for histological analysis of brain tissue of rats with HIE model¹. We observed the characteristic of HIE, including neuronal damage and neuronal cellular edema. These observed pathological changes and alterations in neuronal morphology were consistent with the established characteristics of HIE, confirming the success of our model⁵⁻⁷. The reason why the infarction in the Model group was not obvious may be caused by the different layers selected when cutting layers. We have re-adjusted the TTC coloring pictures and marked them with circles.

The phenomenon of bilateral cerebral infarction caused by unilateral carotid artery ligation has been reported in the literature, which may be caused by impaired circulation function of the lateral cerebral branches⁸⁻⁹.

Thank you again for your advice, and we will pay more attention to this kind of problem in future research. In the meantime, we have added this section to the discussion section of the new manuscript (highlighted in red at line 245-248).

References

- [1] Chen X, Song W, Song Y, et al. Lipidomics reveal the cognitive improvement effects of *Acer truncatum* Bunge seed oil on hypoxic-ischemic encephalopathy rats. *Food Funct.* 2023.
- [2] Sampath D, Valdez R, White AM, Raol YH. Anticonvulsant effect of flupirtine in an animal model of neonatal hypoxic-ischemic encephalopathy. *Neuropharmacology.* 2017. 123: 126-135.
- [3] Huang J, Lu W, Doycheva DM, et al. IRE1 α inhibition attenuates neuronal pyroptosis via miR-125/NLRP1 pathway in a neonatal hypoxic-ischemic encephalopathy rat model. *J Neuroinflammation.* 2020. 17(1): 152.

- [4] Niu RZ, Xiong LL, Zhou HL, et al. Scutellarin ameliorates neonatal hypoxic-ischemic encephalopathy associated with GAP43-dependent signaling pathway. *Chin Med*. 2021. 16(1): 105.
- [5] Zheng Y, Li L, Chen B, et al. Chlorogenic acid exerts neuroprotective effect against hypoxia-ischemia brain injury in neonatal rats by activating Sirt1 to regulate the Nrf2-NF-κB signaling pathway. *Cell Commun Signal*. 2022. 20(1): 84.
- [6] Zhou Y, Wang S, Zhao J, Fang P. Asiaticoside attenuates neonatal hypoxic-ischemic brain damage through inhibiting TLR4/NF-κB/STAT3 pathway. *Ann Transl Med*. 2020. 8(10): 641.
- [7] Cai Y, Li X, Tan X, et al. Vitamin D suppresses ferroptosis and protects against neonatal hypoxic-ischemic encephalopathy by activating the Nrf2/HO-1 pathway. *Transl Pediatr*. 2022. 11(10): 1633-1644.
- [8] Verstraete M. Potential and problems with the clinical use of heparin. *Scand J Haematol Suppl*. 1980. 36: 1-24.
- [9] Vernieri F, Pasqualetti P, Diomedei M, et al. Cerebral hemodynamics in patients with carotid artery occlusion and contralateral moderate or severe internal carotid artery stenosis. *J Neurosurg*. 2001. 94(4): 559-64.

2. The authors stated “In this study, we found ASO administration rats group exhibited reduced cerebral edema compared to the Control group”, but the cerebral edema in the Model group is not obvious.

Dear sir or madam,

Thank you.

To provide a more precise description of the results of Hematoxylin and Eosin (HE) staining, we have revised the term “cerebral edema” to “neuronal cells edema”^{1,2}. Additionally, the description of the section "Abstract" in the manuscript has been revised (highlighted in red at line 4).

We reorganized the arrangement of HE stained sections, and added 400X pathological sections to Figure 1b and Supplementary Figure 1b of the new manuscript, enabling a clearer visualization of the alterations in brain injury.

Reference:

- [1] B. Li, C. Dasgupta, L. Huang, X. Meng and L. Zhang, MiRNA-210 induces microglial activation and regulates microglia-mediated neuroinflammation in neonatal hypoxic-ischemic encephalopathy, *Cell Mol Immunol*, 2020, 17, 976-991.
- [2] J. Huang, W. Liu, D. M. Doycheva, M. Gamdzyk, W. Lu, J. Tang and J. H. Zhang, Ghrelin attenuates oxidative stress and neuronal apoptosis via GHSR-1alpha/AMPK/Sirt1/PGC-1alpha/UCP2 pathway in a rat model of neonatal HIE, *Free Radic Biol Med*, 2019, 141, 322-337.

3. In line 276-279, the authors should give more details about the method for HE staining. How many objectives did authors selected? How many fields were quantitated? How was the statistical significance achieved?

Dear sir or madam,

Thank you. We have revised the title of Figure 1b and Supplementary Figure 1b to “HE staining showed histological changes in each group, and all three replicates had similar staining results”.

In addition, the results of three repeated stained samples were selected for statistical analysis, and four domains (1002 × 1008 pixels) were selected for each sample for quantitative analysis. To correct this, the description of the "HE staining" section in "Materials and Methods" in the new manuscript has been modified (highlighted in red at line 293-294).

4. The authors stated “The results of the open field test displayed that the performance of distance moved and duration in center in ASO group was significantly more effective than the Control group, suggesting that the autonomic motor

ability of mice was significantly improved after taking ASO” but there was a deduction in distance moved and duration in center in ASO group.

Dear sir or madam,

Thank you. We reviewed the original data of the open field experiment and found that there were errors in the mapping process. In the previous Figure 1e-f and Supplementary Figure 1c, "Control" and "ASO" were written in reverse.

In the revised version of the manuscript, we have corrected Figure 1e-f and Supplementary Figure 1c, ensuring that the data is accurate. Additionally, we have included the raw data of the open field experiment in the supplementary materials to provide evidence supporting our conclusion. The open field experiment, presented as a scatter plot, clearly demonstrates that the ASO group exhibited significantly improved performance in terms of distance moved and duration in the center compared to the Control group.

5. Supplementary figures 1a, 1c, 1d, and figure 3d were not described in results section.

Dear sir or madam,

Thank you. According to your comments, we have included the descriptions of Figures 1a, 1c, and 1d in the ASO improved autonomic motor ability in HIE rats. Additionally, we have added a description of Figure 3d to the section on the transcriptome analysis of brains in both the ASO and Control groups. The specific modifications are as follows.

The damaged areas in the figure were marked, and the results showed that the volume ratio of the ischemic area stained by TTC in the Model group was 12.84% (Supplementary Fig. 1a) (highlighted in red at line 49-51). The movement duration in the open field test was measured to compare the locomotor activity between the two groups. We found that there were no significant differences in movement duration between Control group and ASO group (Supplementary Fig. 1c) (highlighted in red at line 54-56). HE staining showed that the nerve cells in the brain tissue of the Sham group were neatly arranged, the nuclei were visible, the nucleoli were clear, and the cytoplasm was abundant, whereas in the Model group, nerve cells were disordered and neuronal edema was observed.

We constructed a graphical representation of the correlation network among 12 genes ($P < 0.05$). Positive correlations between genes were depicted in red, while negative correlations were indicated in blue (Fig. 3d) (highlighted in red at line 97-100).

In the new manuscript, we ensured that these figures are appropriately described and discussed in the results section.

6. The descriptions of figures 2d, 2e and 4j in the text is not clear.

Dear sir or madam,

Thank you. The enrichment analysis indicated that the Phagosome pathway exhibited the highest degree of enrichment in the brain, followed by Systemic lupus erythematosus, Staphylococcus aureus infection, Leishmaniasis, Tuberculosis, and other pathways (Fig. 2d). The Phagosome pathway is associated with lipid metabolism, signal transduction, and immune

response. Antigen processing and presentation are related to lipid metabolism, redox homeostasis, and immune response. Based on GO analysis, the results indicate that the membrane of cellular components (GO:0016020 and GO:0016021) exhibited the highest bar height, suggesting the most significant enrichment of this functional category in the gene set (Fig. 2e) (highlighted in red at line 73-79). The present study elucidated the lipid metabolism transformation pathway in rats following ASO administration during hypoxia-induced erythropoiesis (HIE) at high-altitude, revealing the active involvement of glycerol phospholipid and sphingolipid metabolism in this process (Fig. 4j) (highlighted in red at line 137-140).

The description of the section "Transcriptome analysis of brains in both ASO and Control groups" and "Untargeted lipid profiling in brain between ASO and Control groups" in the new manuscript has been revised.

7. In line 126-129, the authors stated phosphatidylglycerols (PG) and phosphatidic acids (PA) significantly increased in the ASO group, which is not consistent with figure 4d.

Dear sir or madam,

We thank the reviewers for their valuable comments, and we amend the description of the manuscript to ensure consistency with Figure 4d. Within the category of glycerophospholipids, the ASO group demonstrated a notable increase in the levels of phosphatidylcholines (PC), phosphatidylethanolamines (PE), phosphatidylserines (PS), and phosphatidylinositols (PI) compared to the Control group. Conversely, the levels of phosphatidyl glycerol (PG) and phosphatidic acid (PA) exhibited a decrease (Fig. 4d-j) (highlighted in red at line 129-132).

The description of the section "Untargeted lipid profiling in brain between ASO and Control groups" in the new manuscript has been revised.

8. The description in line 146-149 is not consistent with figure 5k.

Dear sir or madam,

Thank you. We investigated the correlation between three metabolites and genes, and the results showed that PI (27:2) + 2O and PI (30:3) + 3O were negatively correlated with *Dhfr*, *Abca12*, *Acsm5*, *Asf1b*, and *RGD1564801*. However, they were negatively correlated with *Zbtb37*. Additionally, *Lpcat1* and *RGD1564801* showed a positive correlation with trans-2-Enoyl-OPC4-COA. *Zbtb37* exhibited a negative correlation with trans-2-Enoyl-OPC4-COA (Fig. 5k) (highlighted in red at 150-153).

The description of the section "Combined pathway analysis with transcriptome and lipidome" in the new manuscript has been revised.

Reviewer #2 (Remarks to the Author):

This article is very interesting, but there are some issues that need to be addressed:

1. Figure 1: The colors in the HE-stained images in Figure 1 and Supplementary Figure 1 are inconsistent. Please check. Additionally, for the quantitative bar graphs, it is recommended to use scatter plots with raw data.

Dear sir or madam,

Thank you. Aiming at the problem of the inconsistent color of HE staining, we selected a new image for replacement, which ensured that the quality of the new image was better. The quantitative bars have been modified as suggested and replaced in a new manuscript (Fig. 1b and d, Supplementary Fig. 1b and d). During the quantitative analysis process, we utilized ImageJ to perform decolorization before calculating the stained area rather than the color depth of the stain. Thank you again for your advice.

2. In Supplementary Figure 1, for the TTC-stained images, the Control group underwent HI modeling, so why does the image not show infarction? Also, quantitative analysis of TTC staining is needed. Additionally, why were only four groups analyzed for TTC, while other analyses were done for only two groups?

Dear sir or madam,

Thank you. We adopted a very mature modeling method to establish a HIE rat model¹⁻³. Although not prominently visible, we have indicated the site of damage in supplementary Figure 1a. Furthermore, in our previous study, we reported the results of HE staining and metabolic analysis in rats undergoing sham surgery and HIE modeling, both of which confirmed the success of the model⁴.

By quantitative analysis using ImageJ, we calculated that the volume ratio of the ischemic area in the TTC stained Model group was 12.84%. We have also revised this description in the section of "ASO improved autonomic motor ability in HIE rats". The volume ratio of ischemic areas was calculated by dividing the sum of the white ischemic areas in each slice by the sum of the brain slices in each slice and multiplying by 100%. Based on our results, it can be concluded that cerebral infarction occurred in the Model group. The reason why the infarction in the Model group of Supplementary Figure 1a was not obvious may be caused by the different layers selected when cutting layers. To address this, we have readjusted the TTC staining images and marked them accordingly.

In our study, we initially established four groups: Sham operation group, Model group, and Control group and ASO group. In our previous study⁴, we compared the Sham group with the Model group and investigated the role of ASO in non-high-altitude regions. This study was carried out based on our previous research and focused on exploring the intervention effects of ASO on HIE in high-altitude areas, as well as the synergistic changes in gene expression and metabolites. Thus, this study primarily compares the data analysis results between the Control group and the ASO group in a high-altitude environment.

References

- [1] Sampath D, Valdez R, White AM, Raol YH. Anticonvulsant effect of flupirtine in an animal model of neonatal hypoxic-ischemic encephalopathy. *Neuropharmacology*. 2017. 123: 126-135.
- [2] Huang J, Lu W, Doycheva DM, et al. IRE1 α inhibition attenuates neuronal pyroptosis via miR-125/NLRP1 pathway in a neonatal hypoxic-ischemic encephalopathy rat model. *J Neuroinflammation*. 2020. 17(1): 152.
- [3] Niu RZ, Xiong LL, Zhou HL, et al. Scutellarin ameliorates neonatal hypoxic-ischemic encephalopathy associated with GAP43-dependent signaling pathway. *Chin Med*. 2021. 16(1): 105.
- [4] Chen X, Song W, Song Y, et al. Lipidomics reveal the cognitive improvement effects of *Acer truncatum* Bunge seed oil on hypoxic-ischemic encephalopathy rats. *Food Funct*. 2023.

3. Figures 2-6: The image in Supplementary Figure 2 is not clear, and the text in the image is blurry.

Dear sir or madam,

Thank you. We have addressed this concern by revising and improving the quality of the image in the revised version of our new manuscript. In order to solve the problem of unclear pictures, we have re-uploaded all the Figures in PDF format, hoping to help readers better understand the figures and the manuscript.

4. Line 73: "Based on GO analysis, membrane of cellular component (GO:0016020 and GO:0016021) was activated in molecular function(Fig. 2e)." This description does not match the displayed image. Please verify.

Dear sir or madam,

Thank you. The description of Fig. 2e was changed to "Based on GO analysis, the membrane of cellular components (GO:0016020 and GO:0016021) exhibited the highest bar height, suggesting the most significant enrichment of this functional category in the gene set (Fig. 2e)". The description of the section "Transcriptome analysis of brains in both ASO and Control groups" in our new manuscript has been revised (highlighted in red at line 73-79).

5. Figure 2 and Figure 6 does not have asterisks (*) indicating statistical significance, so the explanations (*, **, *) in the legend are not needed.**

Dear sir or madam,

Thank you. According to your suggestion, the description of "*/**/**" indicates P -value < 0.05 / < 0.01 / < 0.001 has been removed from the legend of Figure 2 and Figure 6.

6. Line 127: "glycerophospholipids (GP), lysophospholipids, plasmalogen, and other lipids (Fig. 4D-G)." The format for Figure 4D-G is inconsistent. Besides, Line 220: (Supplementary Figure 2H) shows inconsistent expression.

Dear sir or madam,

Thank you. We have fixed all similar errors in our new manuscript. "Fig. 4D-G" has been replaced by "Fig. 4d-g" and "Supplementary Figure 2H" has been replaced by "Supplementary Figure 2h" (highlighted in red at line 129).

7. Line 146: "We investigated the correlation between three metabolites and genes, and the results showed that PI (27:2)+2O and PI (30:3)+3O were positively correlated with Taappc2, Spcs3, Naf1, Napg, Rab7a, Brk1, Atp6v1c1,

Rexo2, Fabp3, and Kansl2, and negatively correlated with Dhfr. There was a positive correlation between Lpcat1 and trans-2-Enoyl149 OPC4-COA, but a negative correlation between Trappc2 and trans-2-Enoyl-OPC4-COA (Fig. 5k)." Please check the description, as it does not match the displayed image.

Dear sir or madam,

Thank you. According to your comments, the description of the section "Combined pathway analysis with transcriptome and lipidome" in the manuscript has been revised (highlighted in red at 150-153).

Specific modifications are as follows: We investigated the correlation between three metabolites and genes, and the results showed that PI (27:2) + 2O and PI (30:3) + 3O were negatively correlated with *Dhfr*, *Abca12*, *Acsm5*, *Asf1b*, and *RGD1564801*. However, they were negatively correlated with *Zbtb37*. Additionally, *Lpcat1* and *RGD1564801* showed a positive correlation with trans-2-Enoyl-OPC4-COA. *Zbtb37* exhibited a negative correlation with trans-2-Enoyl-OPC4-COA (Fig. 5k).

8. The Discussion section should avoid excessive repetition of the results.

Dear sir or madam,

Thank you. Based on your suggestion, we have removed unnecessary descriptions of the results from the discussion section in the new manuscript (highlighted in red at line 249-253). We hope these changes improve the overall clarity and flow of our new manuscript.

9. In the Methods section, please provide the total number of animals used, as well as the number in each group, including sham and model groups.

Dear sir or madam,

Thank you. After successful modeling for 48 h, the HIE rats were reclassified into two groups: Control group (n=35) and ASO group (n=35). After 30 days of continuous gavage intervention, eight rats in each group were randomly selected for behavioral evaluation. Unfortunately, during the experiment, two rats in the Control group and one rat in the ASO group died. In addition, we collected separately all remaining rat brain tissue from each of the two groups for histological evaluation (n=6), lipidomics (n=8), and transcriptomic studies (n=13). The number and analysis of Sham group (n=8) and Model group (n=8) were in our other study. The description of the section "Experimental animals and treatments" in the new manuscript has been revised (highlighted in red at line 264-269).

10. In the Transcriptomic analysis and Lipidomics analysis, please provide the number of samples tested in each group and specify the specific brain regions. Additionally, it would be beneficial to include validation of key genes or lipids if possible.

Dear sir or madam,

Thank you. Throughout the study, we randomly selected eight mice per group for lipidomics and 13 rats per group for transcriptomics. We performed the tests using whole brain tissue. Since the nerve fibers in the brain are mainly concentrated in the white matter, the outer myelin sheath membrane of nerve fibers harbors nearly half of the brain's lipids¹. Myelin itself comprises approximately 75–80% lipid by dry weight, with cholesterol, phosphatidylcholine, sphingomyelin, ceramide, glucosyl-ceramide, and sulfatide being the major lipid species². In adult white matter, NA constitutes 36% of the major fatty acid component in sphingomyelin^{1,3}. Unlike DHA mainly in the cerebral cortex, the lipids involved in ASO including Linoleic acid (LA, C18:2 ω -6, 30.7%), Oleic acid (OA, C18:1 ω -9, 21.8%), Erucic acid (EA, C22:1 ω -9, 18.7%), Eicosenoic acid (CEA, C20:1 ω -9, 8.54%), and Nervonic acid (NA, C24:1 ω -9, 6.89%), are distributed in multiple brain regions *in vivo*.

In fact, after the performance of transcriptomic and lipidomic assays, there were not enough sample materials to carry out a complete validation measurement for a large number of biomarkers, such as gene expressions or lipid concentrations. Compensating for that, we presented some upstream and downstream relationships between metabolites and gene expression, which can be correlated and validated against each other. For example, in the process from FAs to FA-CoA (green layout), up-regulation of the thromboxane A synthase 1 gene (*Tbxas1*, Fig. 6a) advanced the transformation of FAs to prostaglandin²⁰, while the increased expression of acyl-coenzyme A synthetase gene (*Acs1*, Fig. 6a) also promoted the transformation of FAs to FA-CoA²¹, and further utilization of structural FAs synthesis.

As you referred, our future studies will focus on the mechanism of ASO-mediated lipid peroxidation in HIE, and at that time the expressions of genes, proteins and metabolites concentrations of our target pathway will be calculated and validated. Thanks again for your advice.

These explanations have been added to the last paragraph of the discussion in our new manuscript. (highlighted in red at line 264-269)

References

- [1] Hayes CE, Ntambi JM. Multiple Sclerosis: Lipids, Lymphocytes, and Vitamin D. *Immunometabolism*. 2020. 2(3).
- [2] Aggarwal S, Yurlova L, Simons M. Central nervous system myelin: structure, synthesis and assembly. *Trends Cell Biol*. 2011. 21(10): 585-93.
- [3] Sargent JR, Coupland K, Wilson R. Nervonic acid and demyelinating disease. *Med Hypotheses*. 1994. 42(4): 237-42.

Reviewers' comments:

Reviewer #2 (Remarks to the Author):

The author has made revisions based on the review comments, but there are still some areas that need further modification before acceptance:

1. In line 54, "The movement duration in the open field test was measured to compare the locomotor activity between the two groups." This sentence is ambiguous. Which two groups are specifically referred to? It's not clear which figure or description is being referenced.

2. Figure 3f and Figures 4a-c are still unclear; the text inside is quite blurry. It's recommended to ensure a uniform font and font size for all text within the figures, and to make the text as clear as possible.

3. In the Methods section, the author mentions in the supplementary materials that there are Sham and Model groups, but in the section describing animal grouping, only "Control" and "ASO" groups are mentioned. Please verify and clarify.

4. In line 265, "Rats in the ASO group were treated with ASO, while rats in the Control group were given the same volume of corn oil as control vehicle." These two groups should have also undergone HIE modeling. Please describe this clearly, including the timing of drug intervention after modeling, despite the presence of the schematic diagram in Figure 1.

5. For language improvement, it's advised to engage a professional for proofreading.

Please note that these suggestions are intended to help clarify and improve the manuscript before final acceptance.

Response to Referees

Reviewer #2 (Remarks to the Author):

1. In line 54, "The movement duration in the open field test was measured to compare the locomotor activity between the two groups." This sentence is ambiguous. Which two groups are specifically referred to? It's not clear which figure or description is being referenced.

Dear sir or madam,

Thank you. We have addressed the ambiguity in the sentence at line 54 with the following revision: "However, no significant differences in movement duration were observed between the Control group and the ASO group in the open field test ". (Highlighted in red at line 59-60).

2. Figure 3f and Figures 4a-c are still unclear; the text inside is quite blurry. It's recommended to ensure a uniform font and font size for all text within the figures, and to make the text as clear as possible.

Dear sir or madam,

Thank you. We have revised and improved the quality of 'Figure 3f and Figures 4a-c' in our new manuscript. Meanwhile, we re-uploaded new Figure 3 and Figure 4 in PDF format.

3. In the Methods section, the author mentions in the supplementary materials that there are Sham and Model groups, but in the section describing animal grouping, only "Control" and "ASO" groups are mentioned. Please verify and clarify.

Dear sir or madam,

Thank you. In the "Methods" section, we added the grouping descriptions of Sham group and Model group: "Especially, before the ASO intervention, we conducted a detailed examination and comparison of the Sham group (n=8) and the Model group (n=8), which was consistent with our previous study⁶³". (Highlighted in red at line 272-274).

4. In line 265, "Rats in the ASO group were treated with ASO, while rats in the Control group were given the same volume of corn oil as control vehicle." These two groups should have also undergone HIE modeling. Please describe this clearly, including the timing of drug intervention after modeling, despite the presence of the schematic diagram in Figure 1.

Dear sir or madam,

Thanks for your suggestion. In the "Experimental animals and treatments" section of our new manuscript, we have revised the description of modeling and grouping of animals: " HIE rats in the ASO group underwent intragastric ASO treatment, whereas HIE rats in the Control group received an equivalent volume of corn oil as the control vehicle⁶⁴. After a 30-day intervention period, eight rats from each group were subsequently chosen randomly for behavioral evaluation."(Highlighted in red at line 268-270).

5. For language improvement, it's advised to engage a professional for proofreading.

Dear sir or madam,

Thank you. We have enhanced the quality of the manuscript through professional proofreading to ensure linguistic accuracy and fluency. (Revision also highlighted in red)